# Intrinsic luminescence blinking from plasmonic nanojunctions

Wen Chen[1], Philippe Roelli[1,2], Aqeel Ahmed[1], Sachin Verlekar[1], Huatian Hu [3], Karla Banjac [4], Magalí Lingenfelder [4], Tobias J. Kippenberg [2], Giulia Tagliabue [5] & Christophe Galland [1]✉

Plasmonic nanojunctions, consisting of adjacent metal structures with nanometre gaps, can support localised plasmon resonances that boost light matter interactions and concentrate electromagnetic fields at the nanoscale. In this regime, the optical response of the system is governed by poorly understood dynamical phenomena at the frontier between the bulk, molecular and atomic scales. Here, we report ubiquitous spectral fluctuations in the intrinsic light emission from photo-excited gold nanojunctions, which we attribute to the light-induced formation of domain boundaries and quantum-confined emitters inside the noble metal. Our data suggest that photoexcited carriers and gold adatom - molecule interactions play key roles in triggering luminescence blinking. Surprisingly, this internal restructuring of the metal has no measurable impact on the Raman signal and scattering spectrum of the plasmonic cavity. Our findings demonstrate that metal luminescence offers a valuable proxy to investigate atomic fluctuations in plasmonic cavities, complementary to other optical and electrical techniques.

[1] Ecole Polytechnique Fédérale de Lausanne, Laboratory of Quantum and Nano-Optics, Lausanne, Switzerland. [2] Ecole Polytechnique Fédérale de Lausanne, Laboratory of Photonics and Quantum Measurements, Lausanne, Switzerland. [3] The Institute for Advanced Studies, Wuhan University, Wuhan, China. [4] Ecole Polytechnique Fédérale de Lausanne, Max Planck-EPFL Laboratory for Molecular Nanoscience, Lausanne, Switzerland. [5] Ecole Polytechnique Fédérale de Lausanne, Laboratory of Nanoscience for Energy Technologies, Lausanne, Switzerland. ✉email: chris.galland@epfl.ch

Plasmonic nanojunctions formed by ultrathin dielectric spacers between two metals enable reaching the quantum limits of light confinement at visible and near-infrared frequencies, with a growing number of applications in molecular science, nanophotonics, quantum optics, and nanoscale optoelectronics[1]. By inserting molecules or low-dimensional materials in plasmonic nanojunctions, their intrinsic optical, electronic, and vibrational properties can be investigated with unprecedented sensitivity[2–4]. Furthermore, these properties can be modified by leveraging giant values of the Purcell factor[5–8], optomechanical coupling rate[9,10], or vacuum Rabi splitting[11]—values that typically surpass those of dielectric cavities. The generation of photo-excited charge carriers inside the metal can be enhanced by the plasmonic resonance and field enhancement, with potential applications in photo-catalysis[12–14] and nanoscale light sources[7]. Despite progress in developing plasmonic nanojunctions as a universal platform to engineer light–matter interaction at the nanoscale, the realization of their full potential is hindered by a limited understanding of physical processes driven by the tightly confined optical fields at the atomic scale[3,10,15–20]. Moreover, the modification of plasmon damping[21,22] and charge carrier dynamics[23,24] by metal–molecule interfaces and intrinsic grain boundaries[25] can further complicate the understanding of plasmonic nanojunctions.

Illustrating the emerging opportunities in this field, the efficiency of intrinsic light emission from a noble metal under optical or electrical pumping can be enhanced by many orders of magnitudes thanks to the giant Purcell factor provided by plasmonic nanocavities[26–31]. This plasmon-enhanced metal photoluminescence (PL) enables an increasing number of applications in imaging and nano-science[32–34]. Although its underlying principles are still under debate[29,31,35–37], it is generally accepted that both interband and intraband transitions in the noble metal contribute to the radiative recombination of photo-excited carriers, with their relative contributions determined by the bulk band structure[38], the electron–hole pair energy[31], and the degree of spatial confinement[39]. At the meso- to macroscopic scale (~10–100 nm) governing the plasmonic response, the band structure of the metal is bulk-like. In contrast, at the atomic scale, studies of metal clusters and nanoparticles below a few nanometers have shown that quantum confinement leads to bright emission from discrete energy states, as well as from metal–ligand hybrid states[16,33,40,41]. To date, these two domains have been largely considered as separate realms.

In this article, we show that such a distinction should be reconsidered. We discover that the intrinsic light emission from gold plasmonic nanojunctions generally consists of two components: (i) a stable light emission baseline, spectrally following the plasmonic resonances and governed by the bulk metal band structure, and (ii) a contribution from quantum-confined emitters and crystal defects randomly forming and disappearing near the metal surface (Fig. 1a). This latter process, which results in a fluctuating (i.e. blinking) luminescence and is the focus of our study, has its origin at the atomic scale, but is made observable thanks to the Purcell effect provided by the plasmonic modes of the entire junction. The Purcell-enhanced emission from quantum-confined metallic emitters transiently results in sharper linewidths (higher apparent Q factors) and much higher quantum yields compared to the baseline emission. Our findings reveal a phenomenology where luminescence blinking is due to metastable configurations of the atomic lattice, instead of fluctuations in the charge state as observed to date in molecular fluorophores and low-dimensional semiconductors[42,43]. They raise interrogations about the validity of using bulk electronic band structures to model chemical and photochemical interactions at the surface of plasmonic structures. We anticipate that our results will motivate further experimental investigations of optically and electrically induced light emission from plasmonic nanojunctions, with specific attention devoted to metastable and transient states of emission and their relationship with modifications in the carrier relaxation pathways.

## Results

**Blinking of metal PL in single plasmonic nanojunctions**. We fabricated plasmonic nanojunctions following the "nanoparticle-on-mirror" approach[44–47]. Starting from a metallic mirror (with thickness > 70 nm), precise control of the spacer thickness was achieved by self-assembly of a molecular monolayer[48], or by the transfer of a transition metal dichalcogenide monolayer[46], or by the atomic layer deposition of an oxide[47]—or by a combination thereof. Subsequently drop-casting nanoparticles of the desired shape and composition (diameter kept at ~80 nm in the following) resulted in the formation of nanojunctions with well-controlled metal spacing, and tailored optical resonances dominated by the excitation of localized surface plasmons with large field enhancement inside the gap (Fig. 1a). In most cases, we encapsulated the final structures in a thin (~5–10 nm) aluminum oxide layer for improved long-term stability. Full details about sample fabrication and characterization are presented in the Supplementary Methods. Overall, we acquired hundreds of PL time-traces on individual nanojunctions in more than 20 different samples with distinct mirror, spacer, and nanoparticle compositions. In order to simultaneously collect vibrational Raman scattering and elastic Rayleigh scattering and to study the temperature dependence of blinking statistics, we built room temperature and cryogenic multi-functional microscopes for single-particle spectroscopy, as schematically depicted in Fig. 1b. A complete list of fabricated samples and details of the setups are described in Supplementary Table 1 and Methods.

We present first the results from a nanojunction consisting of a chemically synthesized gold flake with (111) surface, a self-assembled biphenyl-4-thiol (BPhT) monolayer, and a commercially available colloidal gold nanoparticle (nominal size 80 nm) (Fig. 1). The plasmonic response of the single nanojunction is first characterized by dark-field (DF) scattering spectroscopy using white light excitation from the side at a glazing angle with tunable polarization, so that specular reflection from the substrate is not collected by the objective lens (Fig. 1b). Without specific mention, the DF measurements in the following are all using p-polarized white light. The DF spectrum exhibits three major features (Fig. 1c). The strong peak in the near-infrared is attributed to a longitudinal dipolar antenna mode (polarized normal to the substrate) with strong field enhancement in the gap (labeled L01 in Fig. 1c). Additionally, when the diameter of the nanoparticle facet in contact with the spacer exceeds about 10 nm, the structure supports Fabry–Pérot-like metal–insulator–metal gap modes. These may hybridize with the vertically polarized antenna modes[49,50] giving rise to higher-order modes labeled S02 (observed around 620 nm) and S11 (overlapping with L01 for this particular nanojunction). Finally, around 530 nm, the transverse plasmon mode of the nanojunction (labeled T) can be observed. These attributions are confirmed by numerical calculations (see Fig. 2g, h) and polarization-dependent DF measurements (Supplementary Fig. 9).

Efficient excitation of metal PL from the single nanojunction is achieved at a wavelength of 532 nm, when the photon energy matches optically allowed interband ($d$ to $sp$ band) transitions in gold. This energy is also resonant with the transverse mode, enhancing the absorption cross-section. The PL emission from a nanojunction is much stronger than the weak continuum PL collected from the bare metal substrate (see Supplementary

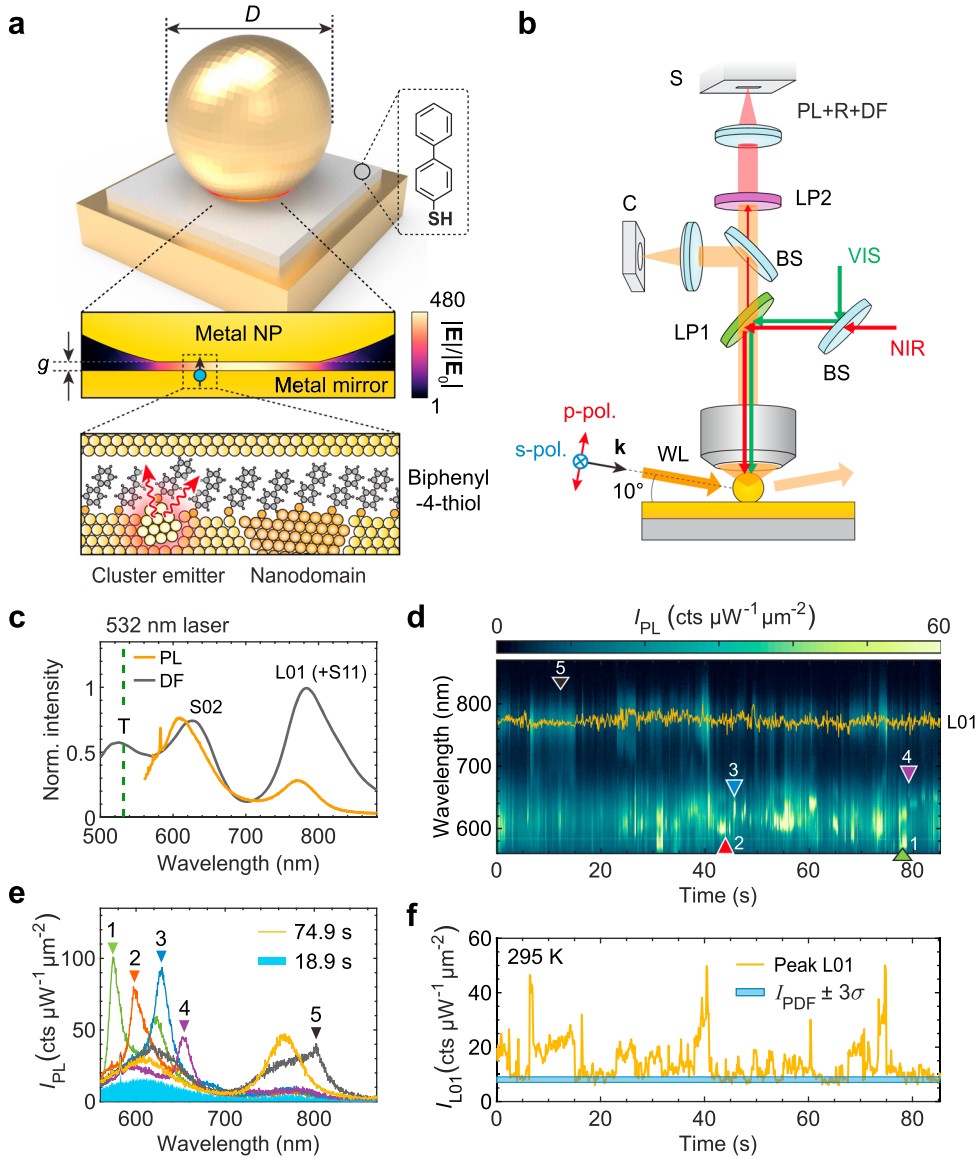

**Fig. 1 Blinking of metal photoluminescence (PL) in a single nanojunction. a** Schematic representation of a nanojunction, made of a gold mirror, a self-assembled biphenyl-4-thiol (BPhT) monolayer ($g \approx 1$ nm), and a faceted gold nanoparticle ($d \approx 80$ nm). Middle inset: simulation of electric field distribution in the nanojunction region. Lower inset: illustration of the luminescent nano-clusters or nano-domains forming under laser irradiation, which we invoke as the cause of PL fluctuations shown in **d**–**f**. **b** Schematic of the optical setup enabling three simultaneous types of measurement: PL under 532 nm (VIS) excitation, Raman scattering (R) under tunable near-infrared (NIR) excitation, and dark-field scattering (DF) under p- or s-polarized grazing angle white light (WL) illumination. BS beam-splitter, LP long-pass filter, C camera, S spectrometer. **c** PL spectrum of a single nanojunction averaged over the entire duration of panel **d** (orange curve) and DF scattering spectrum of the same nanojunction (gray curve, calibrated by the illumination spectrum). Labeling of the modes is detailed in Fig. 2g, h. **d** Time series of plasmon-enhanced PL (the color scale is saturated for better visibility of weak emission periods). Power density of the 532 nm laser: ~70 µW µm$^{-2}$, camera exposure time = 0.1 s, numerical aperture: 0.85, room temperature. **e** Individual examples of anomalous emission (referring to **d**) deviating from the typical baseline PL emission (shaded blue). **f** Time trace of the maximum PL intensity around the L01 mode, as marked by orange trace in **d**. The shaded blue area corresponds to the instrument-limited level of fluctuations, where $I_{PDF}$ is the peak in the intensity probability density function (PDF) and $\sigma$ is the standard deviation of the measurement noise at this signal intensity (see Supplementary Fig. 7).

Fig. 8), despite the fact that the area of the nanojunction is at least 500 times smaller than our spot size. Here, the PL spectrum is the time average of the series shown in Fig. 1d. This demonstrates that PL from the metal is enhanced by orders-of-magnitude due to the combined effect of large near-field coupling to the nanocavity modes and efficient far-field coupling through the antenna effect[8].

When recorded with a short exposure time (0.1 s), the PL time trace of the nanojunction features pronounced blinking and spectral wandering (Fig. 1d, see Supplementary Movie 1 for the entire time trace). Closer inspection of PL spectra at selected times (Fig. 1e) reveals prominent intensity fluctuations of the L01 mode (orange curve) as well as the appearance of randomly occurring bright PL emission lines around the S02 and S11 modes (green, red, blue, purple, and gray curves). At all times, we also observe the presence of a persistent baseline emission, which corresponds to the weakest emission of the time series (blue-shaded areas in Fig. 1e and Fig. 2c). This baseline PL is attributed to Purcell-enhanced radiative recombination of non-thermal excited carriers through both inter- and intraband processes, as

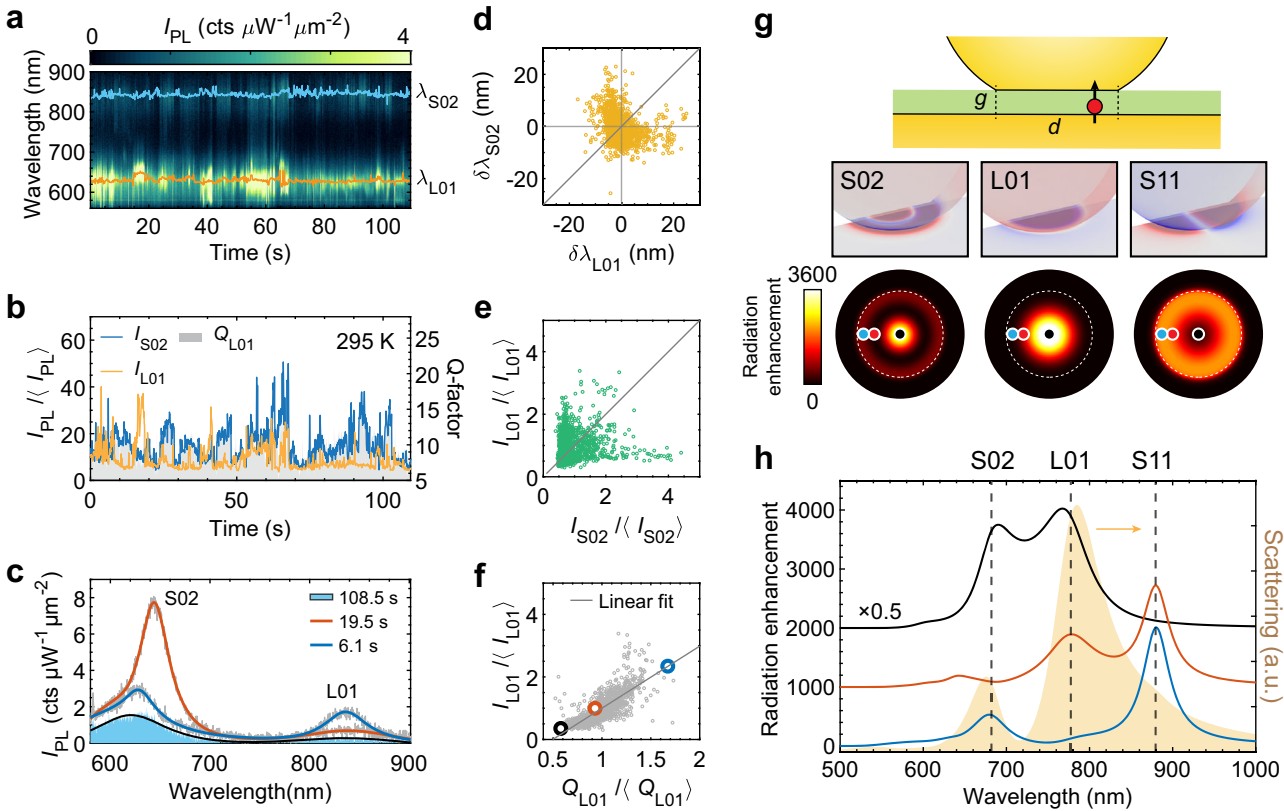

**Fig. 2 Multi-mode blinking: evidence for spatially localized fluctuating sources of emission. a** Fluctuating PL time trace from a nanojunction emitting from the L01 and S02 gap modes. Excitation power density at 532 nm: ~45 µW µm$^{-2}$, numerical aperture 0.85, exposure time: 0.1 s, room temperature. **b** Time series of peak PL intensities at the L01 (blue curve) and S02 (orange curve) resonances along with the $Q$ factor of the L01 mode (gray-shaded area). **c** Examples of PL spectra with different $Q$ factors (as fitted by Lorentzian functions), along with the typical baseline PL spectrum (blue-shaded area). **d, e** Distribution of the PL peak wavelength (**d**) and intensity (**e**) (relative to their time average denoted by brackets) around one resonance vs. the other, showing no correlations. **f** Distribution of relative PL intensity vs. $Q$ factor for the L01 emission, showing a positive correlation. Individual spectra from **c** are highlighted with black, red, and blue circles. **g, h** Full-wave simulation (finite-element modeling) of the optical response of a faceted nanojunction. **g** Based on the surface charge distributions taken at resonance, the modes are identified as the lowest frequency Fabry–Perot-like transverse cavity mode S11, the dipolar bonding antenna mode L01, and the higher-order cavity mode S02, respectively[50]. These modes feature distinct spatial distributions of their local photonic density of states (PDOS), which result in different far-field emission spectra (solid lines in **h**, offset for clarity) when a radiating point dipole is placed at the different locations shown by color-coded full circles in **g**. The yellow-shaded area in **h** corresponds to the calculated DF spectrum.

discussed in previous literature[27,28,31]. The PL peak intensity around the L01 mode (Fig. 1f, orange curve) exhibits prominent fluctuations lasting from milliseconds (see below) up to seconds, well beyond the 3σ interval of the calibrated measurement noise, which includes shot noise and technical noise (Fig. 1f, blue-shaded area; see details in Supplementary Fig. 7). PL blinking was also consistently observed at lower temperatures (see below, and Supplementary Fig. 16).

**Evidence for light-induced fluctuating local emitters**. To gain further insight into the origin of metal PL blinking, we analyze the spectral wandering and lineshape narrowing that accompany blinking, and study the correlations that may exist between fluctuations in emission wavelength, intensity, and linewidth from different regions of the full spectrum. Figure 2a displays another representative time trace with a typical multi-peak PL, with selected spectra shown in Fig. 2c. For each mode, we track the wavelength of maximum PL ($\lambda_{L01}$ and $\lambda_{S02}$), the peak intensity ($I_{L01}$ and $I_{S02}$), and the linewidth (expressed here in terms of $Q$ factor, with $Q_{L01}$ shown as the shaded area in Fig. 2b). A first result of this analysis is that no significant (anti-)correlations exist between the peak wavelengths $\lambda_{L01}$ and $\lambda_{S02}$ (Fig. 2d), nor between the normalized peak intensities $I_{L01}$ and $I_{S02}$ (Fig. 2e).

Any model relying on the modification of the entire, mesoscopic plasmonic response would therefore be difficult to reconcile with our observations.

In contrast, we observe a clear positive correlation between the relative increases of emission intensity ($I_{L01}/\langle I_{L01}\rangle$) vs. $Q$ factor ($Q_{L01}/\langle Q_{L01}\rangle$), Fig. 2f. In other words, the higher the blinking PL intensity, the narrower the effective PL linewidth. From the point of view of traditional mechanisms proposed so far to describe plasmon-enhanced light emission from metal, such behavior is difficult to rationalize. Indeed, an increase in $Q$ factor together with increased radiation rate would reflect a reduction of the nonradiative plasmonic losses, and we are not aware of a mechanism that could lead to such drastic variations over millisecond time scales. All observations suggest instead that an atomic-scale mechanism is causing PL blinking, without affecting the overall plasmonic response.

To check if fluctuating point-like emitters could yield such a behavior, we implemented full-wave simulations of a nanojunction consisting of an 80 nm Au nanoparticle with facet diameter $d = 40$ nm on a Au mirror with spacer thickness $g = 1.3$ nm (Fig. 2g, see details in Supplementary Methods). Under the same illumination and collection geometry as used in the DF measurement, the simulated scattering spectrum (shaded yellow curve in Fig. 2h) matches our experimental data (Fig. 1c). Three

localized gap plasmon modes S02, L01, and S11 can be identified from their distinctive surface charge distributions (middle three panels in Fig. 2g). To emulate a randomly generated point-like emitter, we use a broadband, vertically oriented electric dipole placed on the metal surface at three different positions (blue, red, and black dots in three bottom panels of Fig. 2g). Different radiation enhancements, determined by the local photonic densities of states (PDOS) and radiation angular distribution, are thus probed depending on the overlap between the emitter position and the field distributions of the different gap modes (Fig. 2h). From these simulations, we infer that spatially localized fluctuations in PL quantum yield are consistent with uncorrelated intensity fluctuations in different modes (Fig. 2e). However, this toy model assumes that the PL spectrum is governed by the local PDOS only; it fails to explain the magnitude of wavelength fluctuations (Fig. 2d) and changes in linewidth (Fig. 2f) that we observe in some instances, in particular for thiol-functionalized gold substrates. For other nanojunctions with purely inorganic spacers such as alumina, PL blinking is less pronounced and is not accompanied by noticeable changes in peak wavelength and linewidth (e.g. Supplementary Fig. 10b, c, f). In the following, we focus our attention on the more pronounced blinking features characteristic of organic spacers.

To accommodate our observations, we propose that bright emission centers, consisting of nanoscale metallic domains and/or metal atom clusters, are being formed in the metal surface layer during laser irradiation. Their optical transitions are dominated by quantum-confined electronic states within the $s$–$p$ band of gold[33,40,41] (possibly hybridized with electronic states of the spacer material, in particular through their sulfur atoms). This model is fully consistent with the results shown in Fig. 2a–f: isolated gold clusters or very small nanoparticles[16,33,40,41] are capable of generating PL emission with a wide range of quantum yields, lifetimes, and center wavelengths (covering visible and near-infrared), determined by their size and metal–ligand interaction. In our structures, the plasmonic modes provide a large Purcell enhancement (Fig. 2g, f) which makes the blinking emission predominant close to the plasmonic resonances observed in DF and in the baseline PL. Moreover, if we attribute the brightest emission periods to quantum-confined states in nano-clusters, their linewidths are expected to be narrower than that of the plasmon, as we do observe (Fig. 2c, f). Their emission wavelengths could also be affected by local charging[51] through the DC Stark effect.

A related mechanism is the temporary formation of new grain boundaries and other localized lattice defects, which can scatter electrons and are thus expected to relax wave-vector conservation, leading to a local increase in intraband radiative recombination rate[39]—but without a reduction in emission linewidth nor a shift in emission wavelength. This mechanism explains well the moderate intensity blinking we observe in all constructed nanojunctions, irrespective of their surface chemistry and spacer material (see Supplementary Figs. 10 and 11). For completeness, we mention that the formation of charge-transfer states[22] or surface dipoles[21] has been shown to increase the electron scattering rate, possibly providing an alternative or complementary explanation for PL blinking.

**PL blinking with stable Raman spectrum.** In contrast to predictions from existing models proposed to explain fluctuations in surface-enhanced Raman scattering[10,15,18,19,52] and background emission in plasmonic nanojunctions[20] (a more comprehensive literature review is provided in Supplementary Note 1), we cannot relate the PL blinking to fluctuations of field enhancement inside the gap, nor to changes in the plasmonic response, as we now

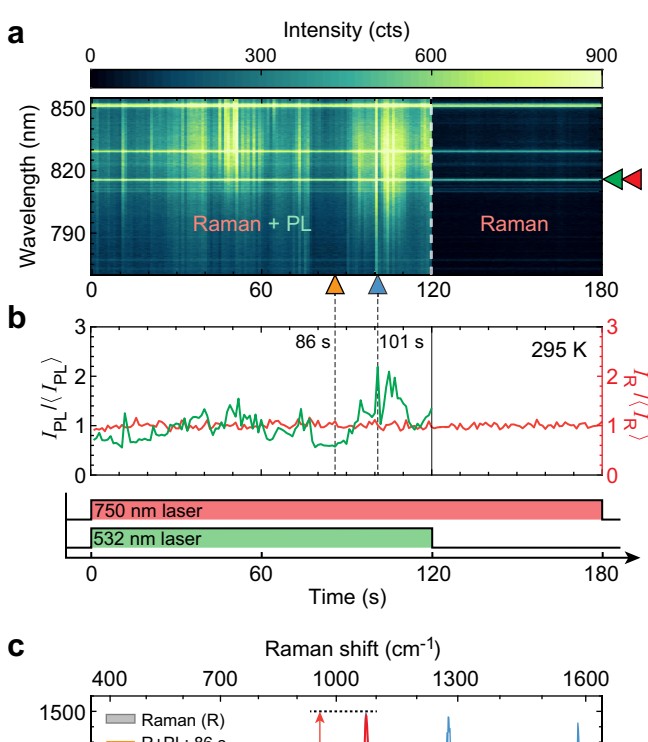

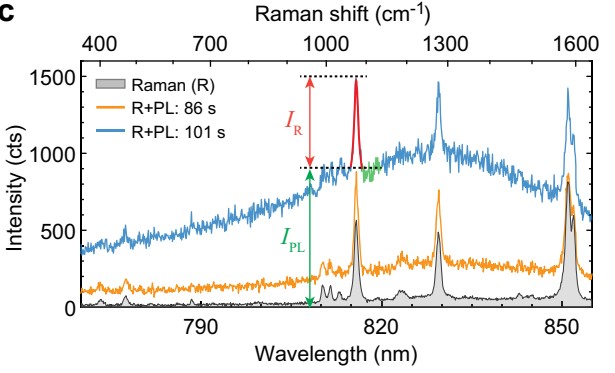

**Fig. 3 Blinking PL with stable plasmon-enhanced Raman spectra. a** Time series of emission spectra acquired under dual excitation with 532 and 750 nm laser beams (first 120 s) with respective power densities ~200 and ~10 μW μm$^{-2}$, and then with 750 nm excitation alone (after 120 s). Camera exposure time = 1 s; numerical aperture: 0.95; room temperature. The color scale is saturated for better visibility of the fluctuations. **b** Time series of the PL intensity (cf. $I_{PL}$ in **c**) and PL-subtracted Raman intensity (cf. $I_R$ in **c**), normalized by their respective time-averages ($\langle I_{PL} \rangle$ and $\langle I_R \rangle$). **c** Example of individual Raman+PL spectra (blue and orange curves) and time-averaged Raman spectrum under 750 nm excitation alone (120–180 s, gray area). The green and red-substituted points on the blue curve highlight how $I_{PL}$ and $I_R$ are defined.

demonstrate. To obtain an independent probe of the local field enhancement, while simultaneously monitoring PL blinking, we performed two-tone excitation with both a 532 nm laser to efficiently generate PL, and with another continuous-wave laser tuned at 750 nm so that the Stokes vibrational Raman signal from the BPhT molecules embedded in the gap is resonant with a near-infrared plasmonic mode (Fig. 3). If blinking were caused by fluctuations in local field enhancement, such fluctuations would be reflected, at least in part, on the Raman signal[10,15,18,46], since molecules are thought to occupy the entire gap region. As a representative example, Fig. 3a shows time series of the Raman +PL (first 120 s) and sole Raman spectra from a nanojunction, with selected Raman+PL and time-averaged Raman (last 60 s) spectra shown in Fig. 3c. Remarkably, the fluctuations of the Raman signal ($I_R$, Fig. 3b, c) remain within the irreducible measurement noise, while much more pronounced fluctuations of

the underlying PL emission ($I_{PL}$, Fig. 3b, c) are observed. This measurement (which was repeated on many nanojunctions with the same result) provides evidence that the near-field enhancement and thereby the local density of photonic states remain stable during PL blinking—in stark contrast with previous observations of fluctuating Raman scattering, e.g. refs. [2,19]. Moreover, based on this observation, we conclude that mechanisms which can be sensitively probed by Raman scattering, including chemisorption[15], adsorbate-metal charge transfer, and charging effects[52], are unlikely to be the dominant cause of PL blinking (see detailed discussion in Supplementary Note 1). Last, if there were a build-up of high DC fields across the gap it should result in a DC Stark shift of the Raman peaks, which we do not observe.

We note that under 750 nm excitation alone, the absence of interband transitions in gold strongly reduces the PL excitation cross-section, and we typically observe a very low amount of PL—except for the brightest blinking events (Supplementary Fig. 13), akin to the so-called "flares" reported in ref. [20]. We also occasionally observe the appearance of many new Raman sidebands, with intensities more than ten times above the normal Raman signal (see Supplementary Fig. 14). Recent reports have invoked the formation of "picocavities"[10,18] to explain such events, which are proposed to be related to metal protuberances causing atomic scale confinement of light. Our measurements show that PL blinks independently of such unusual Raman events, confirming that a new mechanism is at play during PL blinking.

**PL blinking with stable DF scattering spectrum.** Next, we design an experiment to verify that the DF scattering spectrum, which sensitively depends on nanoparticle shape and gap size[1,49], remains stable over time under green light excitation while PL blinks (Fig. 4 and Supplementary Movie 2). Figure 4a shows spectral time series from a single nanojunction under sequential illuminations with 532 nm laser alone; together with white light; and with white light alone. To allow quantitative comparison between the DF and PL fluctuations we plot the probability density functions (PDFs) of the L01-related peak intensity (Fig. 4b) and peak wavelength (Fig. 4c). While the PL features strongly fluctuating intensity and peak wavelength, the elastic scattering of white light is highly stable, even when the laser is simultaneously exciting the nanojunction (PL+DF in Fig. 4a). Therefore, we conclude that rapid changes in nanoparticle shape cannot be the cause of PL blinking. Similarly, the model proposed in[20], which invokes defects in the metal that alter the plasmonic resonance, also fails to agree with our measurements, since it predicts fluctuations of the elastic scattering spectrum correlated with brighter emission. Additionally, the stable plasmonic response in our system also excludes the appearance of quantum-tunneling-induced charge-transfer plasmons, electroluminescence of which was used to explain the broadband fluctuations in ref. [15].

**Dependence of PL blinking on temperature and laser power.** Finally, we turn our attention to the possible mechanisms for the formation and disappearance of the localized emitters responsible for blinking. We measured PL as a function of sample temperature and laser power (Fig. 5). To obtain a larger temporal dynamic range, we employed a single photon counting module (behind suitable filters to select emission from one plasmonic mode, see the inset of Fig. 5a). Figure 5a shows the time series of the emission intensity from a single nanojunction at a sample temperature varying from 4 K up to room temperature, where the measured counts are summed into 1 ms time bins. From the

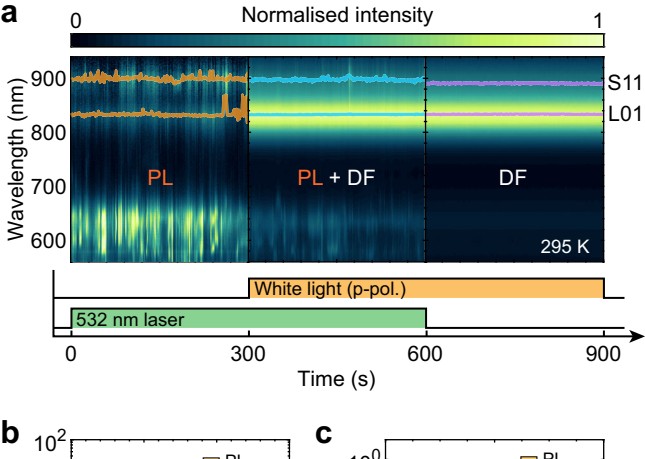

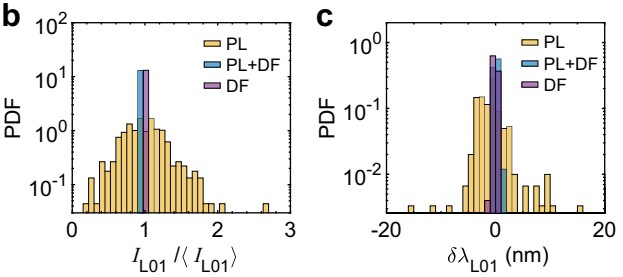

**Fig. 4 Blinking PL with stable dark field. a** Spectral time series from an individual nanojunction under sequential illuminations with 532 nm laser alone (PL), both 532 nm laser and white light (PL+DF, normalized by illumination spectrum) and white light alone (DF, normalized by illumination spectrum). The positions of maximum emission close to the L01 and S11 modes are shown as orange, blue, and purple traces for the PL, PL+DF, and DF regions, respectively. **b, c** Probability density functions (PDFs) for the relative L01 peak intensity (**b**) and wavelength shift (**c**) extracted from the PL (orange), DF (purple), and DF+PL (blue) regions in **a**. The DF+PL (blue) area in **b** are slightly offset away from the value of 1 to show it clearly. Experimental parameters: objective numerical aperture = 0.8; laser power density ~ 45 µW µm$^{-2}$; white light is p-polarized; exposure time = 1 s, room temperature.

enlarged view (Fig. 5b) we clearly identify the stable baseline PL intensity together with much brighter events, many of them lasting for few milliseconds only. Even though Fig. 5a displays more frequent bright events at low temperature, we could not confirm any general relationship between the sample temperature and the blinking statistics in the range of 4–300 K, as illustrated in Supplementary Fig. 16 by measurements performed on a larger number of nanojunctions. Consequently, we can reject the hypothesis that the generation of localized emitters is thermally activated—even though longer-lasting bright events seem more likely to be observed at lower temperature, suggesting that the relaxation to the baseline state may have a thermal component. Multi-physics simulations (see Supplementary Fig. 19) confirm this conclusion by showing a rise in temperature due to laser illumination of a few Kelvin only—negligible compared to the variation of bath temperature explored in Fig. 5a, b.

In Fig. 5c, d, we present the emission statistics as a function of excitation power for a fixed sample temperature (295 K). We find that blinking is hardly observable at the lowest excitation intensities (below ~10 µW µm$^{-2}$, where the stable baseline emission intensity is well above the dark count level with fluctuations barely exceeding the irreducible measurement noise (blue area in Fig. 5c). In contrast, as the laser intensity is increased, PL blinking is activated and becomes more pronounced and frequent, as illustrated by the power-dependent PDFs plotted in Fig. 5c. These observations are confirmed by

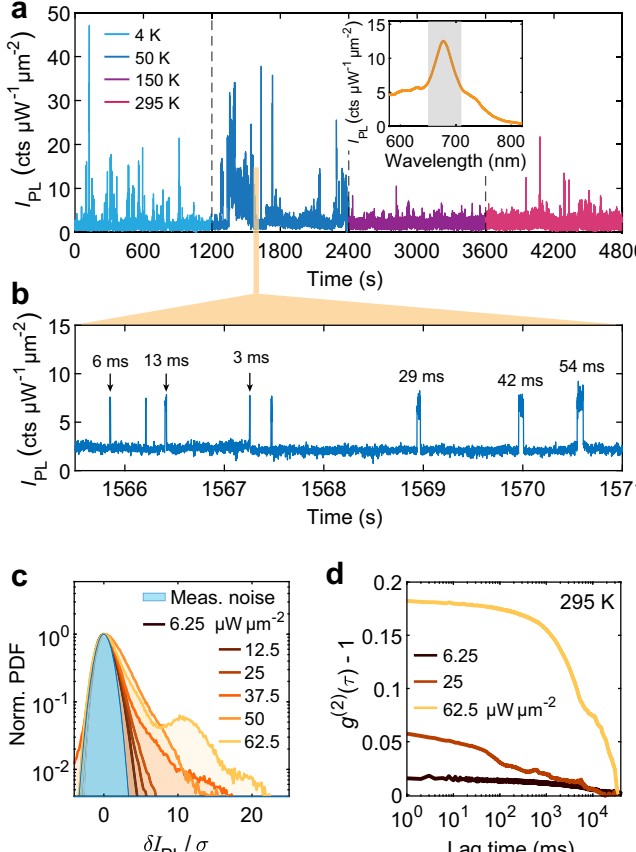

**Fig. 5 PL blinking as a function of laser power and sample temperature.** **a** Temperature-dependent time series of PL intensity from a nanojunction, with the time-averaged PL spectrum shown in the inset. The PL signal was measured by a single photon counting module after spectral filtering (range shown in inset). Binning time, 1 ms. Excitation power density: ~50 $\mu$W $\mu m^{-2}$. **b** enlarged view of **a** revealing shorter blinking events. More data presented in Supplementary Fig. 16 suggest that no clear relationship exists between temperature and blinking statistics. **c** Probability density functions (PDFs) of peak PL intensity as a function of excitation intensity (room temperature) plotted against the re-scaled intensity $\delta I_{PL}/\sigma$ to enable comparison. Here $\delta I_{PL}$ represents the PL intensity deviation from the peak of the PDF; $\sigma$ is the standard deviation of the measurement noise at this signal level. The PDFs are all centered around the averaged PL of the "OFF" state and normalized to the respective measurement noise (blue area). The corresponding time series can be found in Supplementary Fig. 17. **d** Autocorrelation function of the PL intensity trace at different excitation intensities, evidencing increased fluctuations at higher laser powers.

computing the autocorrelation (Fig. 5d) of the PL intensity traces under different laser powers, evidencing a higher level of PL fluctuation at higher laser powers. This result highlights the key role of local optical field strength in activating the localized blinking emitters.

## Discussion

We also find that blinking is much more likely to be activated under 532 nm excitation than at a longer wavelength beyond 600 nm (cf. Supplementary Fig. 13). It is even possible to activate blinking with temporary 532 nm illumination, and see the persisting increased level of background luminescence probed by a near-infrared laser immediately afterward (see Supplementary Fig. 15). While we cannot totally exclude the role of near-field optical forces acting within the nanojunction[17,53], this

observation points to the key contribution of photo-excited electron–hole pairs in inducing the lattice restructuring, since 532 nm is close to the onset of interband absorption in gold.

Previous studies of metals under pulsed laser excitation have demonstrated the existence of a "blast force" due to non-equilibrium electrons, which may deform the metal lattice[54]. On the other hand, it was demonstrated that hot electron injection upon a voltage pulse across a tip-surface junction can result in the restructuring of Au(111) surface in the presence of molecules at cryogenic temperatures[55], which establish a possible link between the existence of hot carriers and the formation of atomic surface defects. More theoretical work should be performed to determine whether similar forces can be relevant under weak continuous-wave excitation of nanoscale plasmonic cavities. Based on simulations (Supplementary Fig. 20) we estimate that on the order of one photon per picosecond is absorbed by the nanojunction under typical excitation powers used here. It is also possible that adatom–molecule complexes may cause localized relaxation centers and hence enhance the probability for a local transfer of energy between non-thermal carriers and the lattice—via a mechanism that could share similarities with electromigration induced by DC currents[56]. In this way, an energy as large as 2.4 eV per photon may be transferred to the lattice on the atomic scale.

Before concluding, we emphasize that blinking appears to be general; it could be observed consistently in many different samples. We investigated the impact of nanojunction composition on blinking by fabricating and characterizing more than 20 different types of nanojunctions, as summarized in Table S1. We systematically changed the substrate type, the spacer layer, and the nanoparticle material and shape, while maintaining similar plasmonic resonance frequencies and mode volumes. A general conclusion can be drawn from these measurements (see Supplementary Figs. 10 and 11): while the molecules alone are not the source of PL, the magnitude and prevalence of PL blinking are indeed influenced by the spacer material and metal surface chemistry (in particular on the substrate side), with molecular spacer yielding more prominent blinking. It could happen for at least two reasons: first, the stability and mobility of surface metal atoms depend on their direct environment, with molecular groups such as thiols perturbing the atomic arrangement in their vicinity and possibly facilitating light-induced restructuring. Second, molecules surrounding the metal can alter its local electron density via charge transfer[21,22], which could favor electron-lattice scattering near the surface. Finally, we characterize the optical response of single Au nanoparticles on $SiO_2$ (Supplementary Fig. 12). In this case, a weak PL blinking event was observed, but the occurrence is even rarer than that found in almost all the nanojunctions. It demonstrates that the existence of a gap mode with a strong local field is essential to the emergence of PL blinking and its observation.

In conclusion, we investigated the intrinsic PL blinking from plasmonic nanojunctions with various compositions, and obtained new insights into the origin of this phenomenon. PL blinking is activated by the excitation laser and persists from room temperature down to 4 K. Bright PL events last from milliseconds up to minutes at low temperature. They can feature linewidths sharper than the plasmon resonance, and wandering peak wavelengths. This behavior contrasts with the nanojunction's baseline emission, its plasmonic response, and local field enhancement, all of which remain stable while PL blinks. These observations can be well explained by the proposed model: metastable localized quantum-confined emitters are photo-induced near the metal surface, and the fluctuating emission is enhanced by near-field coupling to the plasmonic antenna modes. The energy responsible for this lattice restructuring is not of

thermal or ohmic origin; instead, we think it is deposited during the ultrafast relaxation of non-thermal photo-excited carriers. This rich physics occurs under weak continuous-wave excitation at tens of microwatts incoming power only, demonstrating the dramatic effect of plasmonic confinement on carrier and lattice dynamics in nanojunctions. Our results promote the studies of plasmonic nanojunctions in a poorly understood regime, involving phenomena at the interface between the atomic and mesoscopic scale. Moreover, our findings raise questions regarding the microscopic mechanisms governing light emission from plasmonic nanojunctions, impacting their applications as nanoscale emitters. Finally, our work demonstrates that gap plasmons form the basis for new classes of materials whose optoelectronic properties are strongly modified by atomic-scale phenomena driven by non-thermal carriers.

## Data availability

The data that support the findings of this study are available in Zenodo repository with the digital object identifier (DOI): https://doi.org/10.5281/zenodo.4533192 (https://zenodo.org/record/4533192).

## Code availability

The codes used to analyze and plot the data presented in this study are available from the authors upon reasonable request.

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

## Acknowledgements
This works was funded by the Swiss National Science Foundation (SNSF) (project number PP00P2-170684), the European Research Council's (ERC) Horizon 2020 research and innovation programme under QTONE (grant agreement No. 820196), and the European Union H2020 research and innovation programme under THOR (grant agreement No. 829067). The authors acknowledge Hongxing Xu for valuable comments and the IPHYS Characterization Platform at EPFL for assistance. P.R. acknowledges support from the Max Planck-EPFL Center for Molecular Nanoscience and Technology and from the European Research Council (ERC) under the European Union H2020 research and innovation programme (grant agreement no. 732894).

## Author contributions
W.C. and C.G. conceived the study; W.C., P.R., A.A., and S.V. performed the experiments; W.C., P.R., and S.V. analyzed the data; H.H. performed the simulations; W.C., P.R., G.T., and C.G. wrote the manuscript. T.J.K. contributed to early ideas that led to this study. K.B. performed high-resolution surface studies. K.B. and M.L analyzed the results and proposed mechanisms based on local surface reconstruction.

## Competing interests
The authors declare no competing interests.
