## [Peer Review File · Nature Communications]

REVIEWER COMMENTS

Reviewer #1 (Remarks to the Author):

The authors report the occurrence of luminescence blinking in plasmonic junctions, a blinking that they associate with the temporary formation of light-emitting centres (e.g. due to defects) as a result of the optical illumination. The reported effect is novel and interesting, definitely enough to ensure publication in Nature Communications. I found the manuscript well-written, describing very systematic work, with detailed descriptions of every experimental set-up and step. All in all, I think the authors have done an excellent job to disprove various possibilities, and I find their argumentation about gap restructuring, and how it would be evidenced in Raman and dark field, quite convincing. At the same time, I am still not convinced that the proposed explanation is the only possible one.

As I mentioned above, the authors have excluded all the different permanent restructuring (welding, faceting e.g.) usually studied by the Baumberg group, because their signature is a change in the dark-field and Raman spectra, which makes perfect sense.

Next, with the same arguments, they have excluded any quantum effects in the ultranarrow junctions. I agree that this is the case for direct tunnelling or charge transfer through the spacer, but I wonder if surface-enabled Landau damping (see works by Khurgin or Mortensen) could be relevant.

What I am also not sure about, is whether the spacers can also be excluded (as possible templates for deformations and the creation of local emitters). As far as I understand, the main argument here is that the observed blinking persists regardless of the spacer, which is less strong proof. I wonder if the authors could also study a simple dimer (e.g. two cubes very close to each other) on glass. If the effect comes indeed from the metal, and the plasmonic junction is only needed to enhance it, then a nanoparticle dimer could work equally well, with the advantage that no third material (well, the substrate and possible molecules on the nanoparticles, but these exist in the nanoparticle on mirror set-up as well) would be involved. This particle on a film geometry has as its main advantage the reproducibility of numerous junctions with the same gap, but this is not critical in this case.

I am also slightly worried about the significant (even larger) signal in silver junctions (Fig. S12), because interband transitions in silver are quite far away from the 532 nm excitation. It would be great if the authors could add yet another laser at shorter wavelengths.

In summary, I am already moderately positive about this paper, but it would be great if the authors could provide more proof for their explanation.

Reviewer #2 (Remarks to the Author):

In this manuscript, the authors report the study of blinking photoluminescence (PL) from nanojunctions in metal nanoparticle over metal film structures. The results are interesting, and the manuscript is well written. However, the following comments need be addressed before I can recommend its publication.

1. The PL spectra from the nanojunctions were normalized by the PL from bare gold substrate. The authors say this normalization is "to eliminate the spectral variations related to the electronic structure of the metal". As the authors claim the PL emission from the nanojunction is due to the atomic restructuring of the metal and depends on the metal surface chemistry, the PL emission from these defect-like emitters can be treated not related to the PL from gold substrate. It seems not reasonable to do this normalization.

For the 21 plasmonic nanojunctions listed in Table S1, the samples 12-14 with monolayer MoS₂ in the nanojunctions are different from others. Since monolayer MoS₂ itself is a photoluminescent

material, coupling with the plasmonic nanojunction would reshape the emission spectra of MoS₂ depending on the plasmonic modes. For the MoS₂-involved samples, the spectral normalization by PL spectrum of MoS₂ (Figure S7c) would highlight the enhancement caused by the plasmon mode. However, for other samples, I cannot see the rationality to normalize the PL spectra. Since the collected PL from the nanojunction includes some signals from the metal substrate, the PL signals from the substrate may be subtracted from the measured PL spectra at the nanojunctions.

2. The authors write the PL blinking “weakly depend on the degree of crystallinity of the substrate”. If the blinking PL comes from the lattice restructuring of the metal substrate and depends strongly on the metal surface chemistry, why doesn’t it strongly depend on the crystallinity of the metal substrate? The authors also say “existing defects, grain boundaries may enhance the probability for a local transfer of energy between non-thermal carriers and the lattice”.

3. The authors claim that the metastable localized emitters in the nanojunctions are formed during the ultrafast relaxation of non-thermal photo-excited carriers. Can ultrafast experiments be performed to reveal the dynamics for the generation of these localized nano-emitters? That would help to determine the origin of these emitters.

4. Without more convincing evidence, I suggest the authors to soften their expression about the origin of the blinking PL. Actually, some possibilities cannot be excluded as the authors claim. For example, the authors say “Carbon contamination can also be excluded as a cause for PL blinking, in particular for the measurements in the cryostat, and also because our nanojunctions are capped by an alumina layer”. Neither the measurements in the cryostat nor the alumina capping can exclude the carbon contamination, since all samples are exposed to “dirty” environments during their preparation.

5. Almost the whole paragraph before the section “Results” is used to write the significance of this study. In my opinion, some arguments are too strong, such as “use this photo-induced luminescence blinking as a tool for monitoring atomic scale motion and field induced material restructuring”, “engineering the quantum yield, spectrum and stability of plasmonic nano-emitters, or suppressing the background emission in SERS for improved chemical resolution”. Without clearly revealing the origin of this phenomenon, and being lack of efficient ways to control this phenomenon, those arguments are somewhat exaggerated.

6. The authors compare their PL blinking with the blinking observed in SERS experiments in literature. I suggest focusing this comparison on the SERS continuum, as the fluctuation of SERS signal is a phenomenon different from the PL blinking and may have different origins. Without this restriction, the comparison may cause confusion and even be misleading.

7. In the caption of Figure 1, “Lower inset: illustration of the luminescent nano-clusters or nano-domains forming under optical forces, which we invoke as the cause of PL fluctuations”. The “optical forces” should be clarified.

8. There are some errors in the manuscript, such as “Fig. 2G,F” on page 10 should be “Fig. 2G,H”; the “Spacer” for samples No. 7 and 8 in Table S1 seems not right.

Reviewer #3 (Remarks to the Author):

Chen et al. presented a systematic study on an interesting blinking phenomenon in plasmon-enhanced metal photoluminescence (PL) from well-defined single-nanoparticle-on-film nanocavities. Strictly speaking, the blinking phenomenon itself is not new and can be commonly seen in surface-enhanced Raman scattering (SERS) measurements, where blinking happens to both Raman peaks and broad background (recently identified as metal PL or inelastic scattering). Unfortunately, after more than four decades of SERS studies, the phenomenon is still not fully understood and often ignored due to the complexity of the interface – both molecules and metal atoms are active. Therefore, revisiting this topic with state-of-the-art experimental techniques is highly valuable to the field. Chen et al. carefully examined the blinking of the metal PL with an IMPRESSIVE list of single nanoparticle experiments with different pump power, excitation wavelength, temperature, sample geometries, etc. As far as I can see, these are challenging

experiments performed with three different setups and a dozen of nanocavity configurations. In particular, they monitored plasmonic resonance (dark-field scattering) and molecular Raman simultaneous with metal luminescence through double excitation schemes, and showed that the blinking event in the metal PL is not correlated with the blinking in the molecular Raman or any change in the plasmonic resonance. Blinking events do not show a clear change with the temperature variation from 4 K to 295 K (Fig. S10). These experiments suggest a very subtle photon-activated non-thermal change in the nanocavity that gives rise to luminescence without changing near-field strength and plasmonic resonance. Such a change was not expected. The authors attributed it to the formation of gold clusters, grain boundaries, or local lattice defects, which relax the momentum conservation in intraband transitions through the spatial confinement. The paper is well-written, and the conclusion is supported by elegant and well-presented experimental results. Although the microscopic origin of the blinking is still not fully conclusive to me, their well-documented results have ruled out most explanations that are obvious and provoke a non-trivial mechanism. Therefore, I strongly recommend the publication of the paper on Nature Communications, after addressing the following concerns.

1. Does single-nanoparticle PL blink or only nanojunctions do? Small gold nanosphere might not give enough PL, but gold nanorods should present strong enough PL with well-defined resonances.
2. At a first glance, the paper shares some similarities with recent results from the Baumberg group (C. Carnegie, et al., Nat Commun 11, 682, 2020), probably Ref. 20. However, the results from the authors are rather different, given that the dark-field scattering remains still when the PL blinks (Fig. 4). How do authors rationalize the difference given that an almost identical geometry was used? Do authors observe a similar change in dark-field scattering when the laser power is increased to a certain threshold? A few comments will be helpful to the community.
3. In the conclusion part, the author claimed that the PL blinking from plasmonic nanojunction is a new phenomenon, which to me is an unnecessary statement that brings questions. The fluctuation of the broad background in SERS is not unusual, and the blinking of PL from plasmonic nanostructures has been reported before. For instance, J. Phys. Chem. B 2003, 107, 9989-9993; J. Phys. Chem. C 2011, 115, 28, 13645-13659.
4. The authors proposed that the formation of gold clusters, grain boundaries, or local lattice defects from internal atomic-scale reconstruction is responsible for the metal PL blinking. This explains the rise of the metal PL, but it is not clear why such grain boundary or lattice defect would only last a few milliseconds. Is this timing related to temperature? Does the nanojunction stop blinking after a certain time of exposure?
5. Surface charging effect was proposed to explain the blinking of second-harmonic generation from plasmonic nanostructures a few years ago (Phys. Rev. B 2009, 80, 161407(R)). Could metal PL blinking come from local charging?
6. The positive correlation between the metal blinking PL intensity and the Q-factor in Fig. 2F is a strong indication of a shrinking of effective size (J. Phys. Chem. C 2016, 120, 37, 20806) at intermittence. Lightning rod effect grants smaller size particle (grains) stronger nearfield intensity inside the metal and leads to stronger PL intensity, which actually supports the authors' cluster/nanodomain picture.
7. The "afterglow" of metal PL in Fig. S17 is very intriguing but puzzling. What is the temperature of the measurements? Presumably, this "afterglow" phenomenon should strongly depend on the temperature. A few more comments on this will be helpful.
8. Several important references are missing when authors reviewed the origin of metal PL and its correlation with SERS. For instance, Phys. Rev. Lett. 2015, 115, 067403; Nano Lett. 2015, 15, 4, 2600; ACS Photonics, 2016, 3, 1248; and Nat. Commun., 2017, 8, 14891.
9. The authors divided I_{junction} by I_{film} to remove the bulk metal response and obtain the plasmonic resonance spectrum (Fig. 1C), which was justified in ACS Nano 2012, 6, 11, 10147 and Nat. Commun., 2017, 8, 14891.
10. The polarization of white light for the dark-field scattering measurements in Fig. 1C should be specified because it is relevant to the interpretation of modes. The modes assignment would be much more clear if authors can present results from both polarizations.
11. In Fig. S8, panel B shows the blinking around S02 resonance but panel A and C show L01 resonance, which could bring some confusion. The author should extend the range of display in panel B to include L01.
12. There are a few typos, such as "unsuspected" in abstract, "furtehrmore", and "Figure SS2" in SI.

REPLIES TO REVIEWER COMMENTS

December 22, 2020

Reviewer 1 (Remarks to the Author)

“The authors report the occurrence of luminescence blinking in plasmonic junctions, a blinking that they associate with the temporary formation of light-emitting centres (e.g. due to defects) as a result of the optical illumination. The reported effect is novel and interesting, definitely enough to ensure publication in Nature Communications. I found the manuscript well-written, describing very systematic work, with detailed descriptions of every experimental set-up and step. All in all, I think the authors have done an excellent job to disprove various possibilities, and I find their argumentation about gap restructuring, and how it would be evidenced in Raman and dark field, quite convincing. At the same time, I am still not convinced that the proposed explanation is the only possible one. As I mentioned above, the authors have excluded all the different permanent restructuring (welding, faceting e.g.) usually studied by the Baumberg group, because their signature is a change in the dark-field and Raman spectra, which makes perfect sense.”

Reply: We appreciate the reviewer’s positive assessment of the significance and novelty of our work. We are also grateful for their constructive comments and suggestions, which have helped us further strengthen the study and manuscript.

“Next, with the same arguments, they have excluded any quantum effects in the ultranarrow junctions. I agree that this is the case for direct tunnelling or charge transfer through the spacer, but I wonder if surface-enabled Landau damping (see works by Khurgin or Mortensen) could be relevant.”

Reply: The mechanism associated with Landau damping has been proposed to explain the reduction of the near field enhancement and the spectral broadening of the plasmonic resonance when narrowing the gap of the junction below one nanometer. It was also proposed in relationship with surface induced damping, as observed when changing the molecular coverage of nanoparticles in, e.g., *Sci. Adv.* 2019; 5 : eaav0704. If Landau damping were the cause of PL blinking we should observe a concomitant change in the plasmonic response, both in the far field (dark-field scattering) and in the near-field (Raman scattering). Based on Raman+PL and DF+PL simultaneous measurements we demonstrate that the PL blinking is not correlated

with any discernible change in plasmonic resonance; we therefore exclude Landau damping as a plausible explanation for strong PL blinking.

“What I am also not sure about, is whether the spacers can also be excluded (as possible templates for deformations and the creation of local emitters). As far as I understand, the main argument here is that the observed blinking persists regardless of the spacer, which is less strong proof. I wonder if the authors could also study a simple dimer (e.g. two cubes very close to each other) on glass. If the effect comes indeed from the metal, and the plasmonic junction is only needed to enhance it, then a nanoparticle dimer could work equally well, with the advantage that no third material (well, the substrate and possible molecules on the nanoparticles, but these exist in the nanoparticle on mirror set-up as well) would be involved. This particle on a film geometry has as its main advantage the reproducibility of numerous junctions with the same gap, but this is not critical in this case.”

Reply: We thank the Reviewer for the opportunity to further elaborate on this issue. As discussed in Supplementary Sec. 3.5, the nanojunctions with different types of spacers do in fact show different blinking behaviours; in particular the blinking is significantly reduced in the case of Au film covered by a dielectric layer. It demonstrates an important role for organic molecules in enabling PL blinking upon activation by laser illumination. We speculate that inert and uniform dielectric spacer covering the Au film reduces light-induced atomic mobility, resulting in relatively stable PL. To clarify this point, we added the following sentence on page 9 in the main text:

“For other nanojunctions with purely inorganic spacers such alumina, PL blinking is less pronounced and is not accompanied by noticeable changes in peak wavelength and linewidth (e.g. Fig. S10b, c and f). In this study, we focus our attention on the more pronounced blinking features described above.”

We agree with the reviewer that the measurement on a pure metallic, air-gapped nanojunction would be an interesting control experiment. Fabricating such a sample is, however, a challenging task, on which we are currently working, but would be out of the scope of this article. Indeed, achieving 1 to 2 nm gaps without using a spacer material requires advanced fabrication techniques, or completely different approaches such as mechanically actuated antennas. In our studies, we believe that the nanojunctions with a dielectric spacer constitute a good control sample and evidence the important role of molecule - gold interaction.

Furthermore, we have characterised the optical response of a 150-nm-diameter single Au nanoparticle, with results shown in Fig. S12. While the PL of the individual nanoparticle is more stable than the PL of any nanojunction, we do occasionally observe blinking events, definitely beyond the measurement noise (Fig. S12g). These weaker blinking events may be triggered by the presence of loosely bound citrate molecules around the nanoparticle. This new experiment demonstrates that the existence of a gap mode with a strong local field is essential to the emergence of PL blinking and its observation.

“I am also slightly worried about the significant (even larger) signal in silver junctions (Fig. S12), because interband transitions in silver are quite far away from the 532 nm excitation. It would be great if the authors could add yet another laser at shorter wavelengths.”

Reply: We thank the reviewer for the opportunity to further clarify this question. Actually the Ag nanojunction was demonstrated to be a more complicated system compare with the Au sample due to the high chemical reactivity of Ag. For instance, a reversible Ag-AgO photochemical reaction loop involving nanocluster generation can be observed even on a simple bare Ag film (J. Phys. Chem. B, 108, 2148, (2004); Chem. Phys. Lett., 401, 52 (2005); Nanotechnology, 19, 035706, (2008)), causing pronounced fluctuations in light emission. In parallel, Ag can also exhibit a strong interaction with CO from the environment (Surf. Sci., 238, 192, (1990)), which also has an impact on the blinking effect. These additional effects make the blinking phenomenon in Ag systems easier to observe, which is a reason why most of the blinking phenomenon was observed from Ag system in previous literature. However, it also makes Ag nanojunctions more complicated to analyze compared to Au systems, which is why we focused our main study on gold. Ag nanojunctions need to be independently studied, which lies beyond the scope of our study. Here we only show some results on Ag nanojunctions as a supplementary comparison. A discussion about this issue has been included in Supplementary Section 3.9:

“To place our work into context, it should be emphasized that most SERS blinking observations were reported from Ag systems, which show pronounced chemical reactivity (32,35,44,53–58), in contrast to gold which is a rather inert substrate in normal environments. Indeed, luminescent Ag adatoms can be photochemically generated from a silver oxide system under laser irradiation (35, 44, 53, 54). Combining this effect with oxidation from Ag to Ag oxide in air, a reversible photochemical reaction loop can be realised, resulting in luminescence blinking even from a bare Ag system without any Raman probe (35,55–57). In parallel, Ag can also exhibit a strong interaction with CO from the environment; the Raman signal of carbon contaminants can even be found from fresh Ag films deposited under high vacuum condition (48, 51). All these effects make the blinking phenomenon in Ag systems easier to observe but more complicated to analyze compared to Au systems, which is why we focused our main study on gold.”

“In summary, I am already moderately positive about this paper, but it would be great if the authors could provide more proof for their explanation.”

Reply: We hope that our replies above, the additional experiments and the modifications of the manuscript will convince the Reviewer that our manuscript is suitable for publication in Nature Communications. We would like to thank the Reviewer for their valuable comments and for the time they spent on reviewing our work.

Reviewer 2 (Remarks to the Author)

“In this manuscript, the authors report the study of blinking photoluminescence (PL) from nanojunctions in metal nanoparticle over metal film structures. The results are interesting, and the manuscript is well written. However, the following comments need be addressed before I can recommend its publication.”

Reply: We appreciate the Reviewer’s positive assessment of our work. We are also grateful for their constructive comments and suggestions, which have helped us further strengthen the study and manuscript. Below are our point-by-point responses.

“1. The PL spectra from the nanojunctions were normalized by the PL from bare gold substrate. The authors say this normalization is “to eliminate the spectral variations related to the electronic structure of the metal”. As the authors claim the PL emission from the nanojunction is due to the atomic restructuring of the metal and depends on the metal surface chemistry, the PL emission from these defect-like emitters can be treated not related to the PL from gold substrate. It seems not reasonable to do this normalization. For the 21 plasmonic nanojunctions listed in Table S1, the samples 12-14 with monolayer MoS₂ in the nanojunctions are different from others. Since monolayer MoS₂ itself is a photoluminescent material, coupling with the plasmonic nanojunction would reshape the emission spectra of MoS₂ depending on the plasmonic modes. For the MoS₂-involved samples, the spectral normalization by PL spectrum of MoS₂ (Figure S7c) would highlight the enhancement caused by the plasmon mode. However, for other samples, I cannot see the rationality to normalize the PL spectra. Since the collected PL from the nanojunction includes some signals from the metal substrate, the PL signals from the substrate may be subtracted from the measured PL spectra at the nanojunctions.”

Reply: We agree with the reviewer that the additional blinking PL signal from the metal quantum emitters should not be normalized in this way due to their different origins. Following this comment, all PL spectra from nanojunctions have been reprocessed by only subtracting the PL from the metal substrate. One exception is the PL from the MoS₂ spaced nanojunctions: it has been divided by the PL from monolayer MoS₂ on Au to remove the contribution of excitonic PL (which is quenched to a large extent, but still visible). **This revision has been performed in all figures displaying PL spectra**, unless stated otherwise.

In some instances though, when the PL spectrum from the nanojunction is divided into a ‘baseline’ part and an ‘additional blinking’ part, it is reasonable to use the old normalization method, since the ‘baseline’ emission is thought to originate from the bulk gold band structure. The normalization based on the division was therefore kept in Fig. S8, which can provide a way to estimate the radiative recombination rate enhancement.

“2. The authors write the PL blinking “weakly depend on the degree of crystallinity of the substrate”. If the blinking PL comes from the lattice restructuring of the metal substrate and depends strongly on the metal surface chemistry, why doesn’t it strongly depend on the crys-

tallinity of the metal substrate? The authors also say “existing defects, grain boundaries may enhance the probability for a local transfer of energy between non-thermal carriers and the lattice”.”

Reply: We thank the Reviewer for the opportunity to further clarify this point. To address this question, we performed high-resolution atomic force microscopy (see Fig. S2) and found that the average grain size of template-stripped and directly evaporated Au films used in our experiment are 154 ± 19 and 45 ± 21 nm, respectively. Noteworthy is that the grain size is in both cases larger than the typical lateral size of the nanojunction given by the facet size of the 80 nm nanoparticle. In particular for template-stripped gold, which is the usual substrate for nanoparticle-on-mirror cavities, crystalline domains are significantly larger than particle facet size. It means that, statistically, the nanojunction region is formed over a crystalline surface, with a low probability to find pre-existing grain boundaries.

Based on this new observation, we propose that the metal crystallinity mostly impacts the global plasmonic response, which we demonstrated to be stable (in a certain laser power regime), independently of the blinking effect. Consequently, the temporary formation of metal clusters is dominated by the existence of adatom-molecule complexes, and is activated by the local optical field. The important mechanisms occur at the metal-molecule interface, where molecular groups such as thiol are known to lead to pronounced restructuring and changes in adatom energy.

Thanks to the Reviewer’s comment, we realize that our previous statement about “existing grain boundaries” was leading to confusion. To address this issue, we have revised the discussion on Page 19 in the main text:

“It is also possible that adatom-molecule complexes may cause extremely localised relaxation centers and hence enhance the probability for a local transfer of energy between non-thermal carriers and the lattice – via a mechanism that could share similarities with electromigration induced by DC currents (56).”

“3. The authors claim that the metastable localized emitters in the nanojunctions are formed during the ultrafast relaxation of non-thermal photo-excited carriers. Can ultrafast experiments be performed to reveal the dynamics for the generation of these localized nano-emitters? That would help to determine the origin of these emitters.”

Reply: We agree with the Reviewer that an ultrafast PL experiment can be a promising way to further explore the origin of the PL blinking; yet it remains out of the scope of our study and will require dedicated equipment, both in terms of laser infrastructure and detection devices. We also note that under pulsed excitation, the dominant PL mechanisms and excitation pathways could be modified, leading to further complexity in the interpretation. We look forward to studying this regime in the future.

“4. Without more convincing evidence, I suggest the authors to soften their expression about the origin of the blinking PL. Actually, some possibilities cannot be excluded as the authors claim. For example, the authors say “Carbon contamination can also be excluded as a cause for PL blinking, in particular for the measurements in the cryostat, and also because our nano-junctions are capped by an alumina layer”. Neither the measurements in the cryostat nor the alumina capping can exclude the carbon contamination, since all samples are exposed to “dirty” environments during their preparation.”

Reply: We agree with the Reviewer that we cannot exclude the possibility of carbon contamination, which is ubiquitous and indeed unavoidable. Nevertheless, we believe that our measurements provide convincing evidence that carbon contamination, although possibly present, is not the driving cause for PL blinking. First, carbon contamination is expected to affect all samples to a similar extent; yet, we observe strong differences in blinking prominence depending on the spacer layer, suggesting that the contamination is not the leading cause of blinking. Second, due to the extreme field enhancement and single- or few-molecule sensitivity of Raman spectroscopy in the nanocavities we use, a high level of carbon contamination would most likely lead to new Raman peaks, which we don't observe in correlation with PL blinking. To soften the statement, the main text was modified accordingly on page 12:

“Moreover, based on this observation, we conclude that mechanisms which can be sensitively probed by Raman scattering, including chemisorption (15), adsorbate-metal charge transfer and charging effects (42), are unlikely to be the dominant cause of PL blinking (see detailed discussion in supplementary material Sec. 3.9).”

“5. Almost the whole paragraph before the section “Results” is used to write the significance of this study. In my opinion, some arguments are too strong, such as “use this photo-induced luminescence blinking as a tool for monitoring atomic scale motion and field induced material restructuring”, “engineering the quantum yield, spectrum and stability of plasmonic nano-emitters, or suppressing the background emission in SERS for improved chemical resolution”. Without clearly revealing the origin of this phenomenon, and being lack of efficient ways to control this phenomenon, those arguments are somewhat exaggerated.”

Reply: We thank the Reviewer for their comments, we understand the concern that is raised. It was certainly not our intention to oversell the findings. To soften the argument, we have removed the comments about the control of blinking.

“6. The authors compare their PL blinking with the blinking observed in SERS experiments in literature. I suggest focusing this comparison on the SERS continuum, as the fluctuation of SERS signal is a phenomenon different from the PL blinking and may have different origins. Without this restriction, the comparison may cause confusion and even be misleading.”

Reply: We fully agree. To address this comment, we have reorganised the paragraph and added some sentences to further clarify the relationship and emphasize the difference between the previously observed SERS continuum fluctuations and the present form of PL blinking. The

revised discussion can be found in Supplementary Sec. 3.9 based on emphasising the difference between the PL and SERS continuum.

“7. In the caption of Figure 1, “Lower inset: illustration of the luminescent nano-clusters or nano-domains forming under optical forces, which we invoke as the cause of PL fluctuations”. The “optical forces” should be clarified.”

Reply: We thank the Reviewer for attracting our attention to this phrasing, which was inadvertently left over from a previous version of the manuscript, when we suspected that near-field optical gradient forces could be a driving mechanism for the formation of blinking emitters. However, based on our final results (in particular the strong sensitivity on excitation wavelength), this hypothesis cannot be substantiated, even if it remains a possible mechanism. Therefore, we have corrected ‘optical forces’ in this caption and changed it to ‘laser irradiation’, stressing the fact that blinking is activated by laser light rather than ambient thermal energy.

8. There are some errors in the manuscript, such as “Fig. 2G,F” on page 10 should be “Fig. 2G,H”; the “Spacer” for samples No. 7 and 8 in Table S1 seems not right.

Reply: We thank the Reviewer for their careful and conscientious review. We have corrected the mistypes accordingly.

Reviewer 3 (Remarks to the Author)

“Chen et al. presented a systematic study on an interesting blinking phenomenon in plasmon-enhanced metal photoluminescence (PL) from well-defined single-nanoparticle-on-film nanocavities. Strictly speaking, the blinking phenomenon itself is not new and can be commonly seen in surface-enhanced Raman scattering (SERS) measurements, where blinking happens to both Raman peaks and broad background (recently identified as metal PL or inelastic scattering). Unfortunately, after more than four decades of SERS studies, the phenomenon is still not fully understood and often ignored due to the complexity of the interface – both molecules and metal atoms are active. Therefore, revisiting this topic with state-of-the-art experimental techniques is highly valuable to the field. Chen et al. carefully examined the blinking of the metal PL with an IMPRESSIVE list of single nanoparticle experiments with different pump power, excitation wavelength, temperature, sample geometries, etc.

As far as I can see, these are challenging experiments performed with three different setups and a dozen of nanocavity configurations. In particular, they monitored plasmonic resonance (dark-field scattering) and molecular Raman simultaneous with metal luminescence through double excitation schemes, and showed that the blinking event in the metal PL is not correlated with the blinking in the molecular Raman or any change in the plasmonic resonance. Blinking events do not show a clear change with the temperature variation from 4 K to 295 K (Fig. S10). These experiments suggest a very subtle photon-activated non-thermal change in the nanocavity that gives rise to luminescence without changing near-field strength and plasmonic resonance. Such a change was not expected. The authors attributed it to the formation of gold clusters, grain boundaries, or local lattice defects, which relax the momentum conservation in intraband transitions through the spatial confinement. The paper is well-written, and the conclusion is supported by elegant and well-presented experimental results. Although the microscopic origin of the blinking is still not fully conclusive to me, their well-documented results have ruled out most explanations that are obvious and provoke a non-trivial mechanism. Therefore, I strongly recommend the publication of the paper on Nature Communications, after addressing the following concerns.”

Reply: We appreciate the Reviewer’s clear and concise summary of the novelty, main results and key impact of our work. We have conducted more experiments and analysis to address the remaining questions, and revised the manuscript accordingly. Below are our point-by-point responses.

“1. Does single-nanoparticle PL blink or only nanojunctions do? Small gold nanosphere might not give enough PL, but gold nanorods should present strong enough PL with well-defined resonances.”

Reply: To answer this interesting question, we have performed additional experiments and studied the optical response of an individual 150-nm-diameter Au nanoparticle on a 100-nm-thick SiO₂ layer above a Si substrate, with results shown in Fig. S12, and reproduced below.

Although the PL signal of the AuNP is generally stable, we can still observe short PL blinking events, but their occurrence is even rarer than that of the most stable cases from Au nanojunctions with ALD alumina spacer. These weak blinking events may be enabled by the presence of loosely bound citrate molecules around the nanoparticle. This new experiment demonstrates that the existence of a gap mode with a strong local field is essential to the emergence of PL blinking and its observation, potentially explaining why this phenomenon had not been identified in previous single-particle studies.

“2. At a first glance, the paper shares some similarities with recent results from the Baumberg group (C. Carnegie, et al., Nat Commun 11, 682, 2020), probably Ref. 20. However, the results from the authors are rather different, given that the dark-field scattering remains still when the PL blinks (Fig. 4). How do authors rationalize the difference given that an almost identical geometry was used? Do authors observe a similar change in dark-field scattering when the laser power is increased to a certain threshold? A few comments will be helpful to the community.”

Reply: We thank the reviewer for the opportunity to further elaborate on this issue. First, upon further increase of the power of 532 nm laser, beyond what is reported in the manuscript, we eventually observe an irreversible shift of the plasmonic mode caused by the permanent change of the nanojunction morphology, as reported in existing literature (e.g. Nano Lett. 2016, 16, 9, 5605–5611). PL blinking, however, is observed well before the onset of this irreversible modification, and seems unrelated.

Despite the use of similar geometries, there is one significant difference between our study and the 2020 report by J. Baumberg’s team. Indeed, our work focuses on the PL excited via

interband transitions at 532 nm, which yields much stronger PL than the “flares” occasionally seen under excitation at longer wavelengths (633 nm in Carnegie, et. al. 2020). Of course, it was not a priori obvious that our results would depart so much from the recent report by our colleagues; it makes our manuscript even more interesting in our opinion, and highlights the relevance of the precise excitation wavelength for the activation of PL blinking from gold nanojunctions. The fact the two phenomena cannot share the exact same origin is evidenced by the following observation, already mentioned in the main text: our measurements do not reveal any change in dark field scattering (which tracks the plasmonic resonances) during PL blinking, while the “flares” from Carnegie et al 2020 are accompanied by a change of the entire elastic scattering spectrum of the nanocavity.

In summary, we believe that, while SERS background “flares” and PL blinking share common features (both are interpreted as atomic effects, and do not strongly modify the local field in the gap), they do originate from two independent mechanisms, which will require further research to be fully understood. We have added some comments (marked in blue) in the main text on page 12:

“We note that under 750 nm excitation alone, the absence of interband transitions in gold strongly reduces the PL excitation cross-section, and we typically observe a very low amount of PL – except for the brightest blinking events (Fig. S13), akin to the so-called ‘flares’ reported in (20).”

and also some comments (marked in blue) in Supplementary Sec. 3.9:

“Recently, the variation of bulk plasma frequency induced by local defects on the metal interface was proposed as a mechanism to explain the fluctuating ‘SERS continuum’ under stable Raman signal in plasmonic hot-spots (42). Although this phenomenon seems at first similar to our results from the two-color PL+Raman measurement (Fig. 2), the ‘SERS continuum’ fluctuations in (42) are predicted to be correlated with a pronounced shift of the entire plasmonic resonance spectrum. It is at odds with our observations of stable DF scattering spectrum during blinking. On the other hand, the authors of (42) explain the ‘SERS continuum’ as electronic Raman scattering rather than electron-hole recombination process, which can apply to measurements under near-infrared excitation (Fig. S10c and S10d) but does not represent the dominant interband transition processes at play under green excitation.”

“3. In the conclusion part, the author claimed that the PL blinking from plasmonic nanojunction is a new phenomenon, which to me is an unnecessary statement that brings questions. The fluctuation of the broad background in SERS is not unusual, and the blinking of PL from plasmonic nanostructures has been reported before. For instance, J. Phys. Chem. B 2003, 107, 9989-9993; J. Phys. Chem. C 2011, 115, 28, 13645–13659.”

Reply: To address this perfectly relevant comment, we have revised the criticized sentence in the main text on page 17 as:

“In conclusion, we investigated the intrinsic photoluminescence (PL) blinking from plasmonic nanojunctions with various compositions, and obtained new insights into the origin of this phenomenon.”

Moreover, we added and commented all mentioned references in Supplementary Sec. 3.9.

“4. The authors proposed that the formation of gold clusters, grain boundaries, or local lattice defects from internal atomic-scale reconstruction is responsible for the metal PL blinking. This explains the rise of the metal PL, but it is not clear why such grain boundary or lattice defect would only last a few milliseconds. Is this timing related to temperature? Does the nanojunction stop blinking after a certain time of exposure?”

Reply: We appreciate the opportunity to clarify this point. Actually the blinking can last from a few ms to several minutes (e.g., the bottom panel in Fig. S16). The purpose of Figure 5b is just to show some of the shortest resolvable events. To avoid any possible misunderstanding, we have highlighted long lasting blinking events in Fig. S16a. Even though the activation of blinking does not show a clear temperature dependence, long lasting bright PL is more likely at temperatures below ~ 200 K, suggesting that a thermally activated relaxation pathway exists.

Regarding the second part of the question, based on Fig. S16b we can find that even after several hours of continuous laser illumination the PL was still blinking. Only when entering irreversible changes in plasmonic resonance under too high power do we see a clear change in blinking behavior, but we prefer to focus our report on the stable cavity regime.

“5. Surface charging effect was proposed to explain the blinking of second-harmonic generation from plasmonic nanostructures a few years ago (Phys. Rev. B 2009, 80, 161407(R)). Could metal PL blinking come from local charging?”

Reply: We thank the reviewer for raising an interesting suggestion to help understand the origin of PL blinking. Local charging is a natural possibility, not only from the given reference, but also in analogy with blinking in semiconducting quantum dots. Despite differences in the samples and optical processes being probed, the mechanisms proposed in Phys. Rev. B 2009, 80, 161407(R) relying on local charging of the nanoclusters might provide new insight into PL blinking. We cannot exclude the impact of the local charging effect on the optical properties of the nanoclusters, and it may lead to, e.g., some of the spectral wanderings that we observe. Nevertheless, this model presents a shortcoming, because we would expect a stark-shift of the Raman lines if strong local fields were present, which we don't observe. The exact mechanism that would directly link surface charging to a change in PL intensity in this metallic system also remains unclear. To give credit to this possible mechanism, we have added the proposed reference in the main text on page 10 when discussing the blinking emitters:

“Their emission wavelengths could also be affected by local charging [Phys. Rev. B 2009, 80, 161407(R)] through the DC Stark effect.”

“6. The positive correlation between the metal blinking PL intensity and the Q-factor in Fig. 2F is a strong indication of a shrinking of effective size (J. Phys. Chem. C 2016, 120, 37, 20806) at intermittence. Lightning rod effect grants smaller size particle (grains) stronger nearfield intensity inside the metal and leads to stronger PL intensity, which actually supports the authors’ cluster/nanodomain picture.”

Reply: We thank the Reviewer for the opportunity to further elaborate on this issue. We agree with the reviewer that the stronger lightning rod effect from the reduced effective size can generate stronger near-field intensity leading to stronger PL intensity. Nevertheless, this effect should also enhance the local field intensity in the gap resulting in the change of Raman signal, as seen in the SERS measurement in the reference mentioned by the Reviewer. Therefore, we tend to believe that narrowing of the PL linewidth is more likely caused by the generation of quantum emitters with discrete energy levels, since we don’t observe change in SERS intensity during PL blinking.

“7. The “afterglow” of metal PL in Fig. S17 is very intriguing but puzzling. What is the temperature of the measurements? Presumably, this “afterglow” phenomenon should strongly depend on the temperature. A few more comments on this will be helpful.”

Reply: The temperature of this measurement was 295 K. For clarity, we have added temperature labels in all relative figures. We have not seen significant differences in the blinking by varying the bath temperature, except for the longer lasting events below ~ 200 K. The comments about this experiment have been added in Fig. S15. We hypothesize that the “afterglow” evidences the activation of localised emitting centers by the 532 nm excitation of interband transitions. Once activated/created, these emitting centers can be further excited by the longer wavelength NIR laser. We also find this phenomenon very intriguing and plan to study it in more details in a future work.

“8. Several important references are missing when authors reviewed the origin of metal PL and its correlation with SERS. For instance, Phys. Rev. Lett. 2015, 115, 067403; Nano Lett. 2015, 15, 4, 2600; ACS Photonics, 2016, 3, 1248; and Nat. Commun., 2017, 8, 14891.”

Reply: We thank the Reviewer for mentioning these papers, which are indeed closely related with our discussion. The references have all been cited in the introduction.

“9. The authors divided I junction by I film to remove the bulk metal response and obtain the plasmonic resonance spectrum (Fig. 1C), which was justified in ACS Nano 2012, 6, 11, 10147 and Nat. Commun., 2017, 8, 14891.”

Reply: Based on the comment from Reviewer 2, we realised that it is not proper to normalize the additional blinking PL signal from the metal quantum emitters in this way due to the different origins. Following this comment, all PL spectra from nanojunctions have been reprocessed by only subtracting the PL from the metal substrate. One exception is the PL from

the MoS₂ spaced nanojunctions: it has been divided by the PL from monolayer MoS₂ on Au to remove the contribution of excitonic PL (which is quenched to a large extent, but still visible). This revision has been performed in all the figures displaying PL spectra.

In some instances though, when the PL spectrum from the nanojunction is divided into a 'baseline' part and an 'additional blinking' part, it is reasonable to use the old normalization method, since the 'baseline' emission is thought to originate from the bulk gold band structure. The normalization based on the division was therefore kept in Fig. S8, which can provide a way to estimate the radiative PL enhancement.

“10. The polarization of white light for the dark-field scattering measurements in Fig. 1C should be specified because it is relevant to the interpretation of modes. The modes assignment would be much more clear if authors can present results from both polarizations.”

Reply: We agree with the Reviewer that the polarization of DF should be specified to help identify the plasmonic modes. To address this, the schematics of p- and s-polarised excitation have been added in Fig. 1b and Fig. S4a, with the relative statement revised accordingly in the manuscript. Unless specifically mentioned, p-polarised light was used for the DF experiments. Additionally, we have added Fig. S9 to show the polarization-dependent DF spectra from several nanojunctions in comparison with their PL spectra.

“11. In Fig. S8, panel B shows the blinking around S02 resonance but panel A and C show L01 resonance, which could bring some confusion. The author should extend the range of display in panel B to include L01.”

Reply: Many thanks for the suggestion. The figure has been changed according to the Reviewer's suggestion.

“12. There are a few typos, such as “unsuspected” in abstract, “furtehrmore”, and “Figure SS2” in SI.”

Reply: We thank the Reviewer for their careful and conscientious review. The manuscript has been polished and identified typos have been corrected.

REVIEWERS' COMMENTS

Reviewer #1 (Remarks to the Author):

The authors have addressed all comments by the three referees. They performed additional experiments, and added arguments that help disprove some of the alternative suggestions by the referees. The revised manuscript is also more careful in terms of the novelty claims and the degree to which the authors can be certain about their (still dominant) explanation of the observed blinking. I believe that the paper can be accepted for publication in Nature Communications, and look forward to reading follow-up works triggered by the referee reports.

Reviewer #2 (Remarks to the Author):

The authors have made a good effort to address the questions. However, I would recommend follow minor corrections to be made before the paper is published: 1, It would be easy to recognize the Raman modes if Raman shift unit is added in Fig. 3c and Fig.S14. 2, There is an error in Fig.S10e: the label is doubled and overlapped.

Reviewer #3 (Remarks to the Author):

The authors have fully addressed my questions raised in the initial review. I recommend the publication of the manuscript in its present form.

Nevertheless, I would like to share one minor comment after noticing the added AFM image (Supplementary Fig. S2a) of the template-stripped gold film. It is somewhat unexpected to see so many grain boundaries in a template-stripped gold film. Instead, films produced through this method can be commonly atomically smooth across at least tens of microns (for instance, Nat. Commun. 10:5544). The difference might come from the evaporation temperature or silicon wafer surface. But it should not influence the conclusion of the paper, since the authors also performed experiments on single-crystalline gold microplates.

REPLIES TO REVIEWER COMMENTS

Reviewer 1 (Remarks to the Author)

The authors have addressed all comments by the three referees. They performed additional experiments, and added arguments that help disprove some of the alternative suggestions by the referees. The revised manuscript is also more careful in terms of the novelty claims and the degree to which the authors can be certain about their (still dominant) explanation of the observed blinking. I believe that the paper can be accepted for publication in Nature Communications, and look forward to reading follow-up works triggered by the referee reports.”

Reply: We appreciate the Reviewer’s assessment that our work can be accepted for publication in Nature Communications. We are also grateful for their constructive contributions that have helped us further strengthen the study and manuscript.

Reviewer 2 (Remarks to the Author)

“The authors have made a good effort to address the questions. However, I would recommend follow minor corrections to be made before the paper is published: 1, It would be easy to recognize the Raman modes if Raman shift unit is added in Fig. 3c and Fig.S14. 2, There is an error in Fig.S10e: the label is doubled and overlapped.”

Reply: We appreciate the Reviewer’s comments and their constructive contributions that have helped us further strengthen the study and manuscript. For Fig. 3c, an additional axis with unit of ‘Raman shift (cm-1)’ has been added to show the Raman shift more clearly. The error in Fig. S14 has been corrected. We thank the Reviewer for pointing it out.

Reviewer 3 (Remarks to the Author)

“The authors have fully addressed my questions raised in the initial review. I recommend the publication of the manuscript in its present form.”

Reply: We appreciate the Reviewer’s assessment that our work can be published in the present form in Nature Communications. We are also grateful for their constructive contributions that have helped us further strengthen the study and manuscript.

“Nevertheless, I would like to share one minor comment after noticing the added AFM image (Supplementary Fig. S2a) of the template-stripped gold film. It is somewhat unexpected to see so many grain boundaries in a template-stripped gold film. Instead, films produced through this method can be commonly atomically smooth across at least tens of microns (for instance, Nat. Commun. 10:5544). The difference might come from the evaporation temperature or silicon wafer surface. But it should not influence the conclusion of the paper, since the authors also performed experiments on single-crystalline gold microplates. ”

Reply: We thank the Reviewer for the opportunity to further elaborate on this point. We confirm that the template-stripped gold film we used in the manuscript was not fabricated under optimised conditions. Indeed, it is due to the limitation in evaporation rate and temperature from our home-made electron beam evaporation setup. Nevertheless, the average grain size of our template-stripped gold film is still significantly larger than the the bottom facet size of the nanoparticles, meaning that the nanoparticle still has large chance to stay on top of a single grain rather than on the boundary. Also, as the Reviewer rightly mentioned, our conclusion is still solid based on the experiments on single-crystalline gold microplates.